# Prevalence of Taurodontism in Contemporary and Historical Populations from Radom: A Biometric Analysis of Radiological Data

**DOI:** 10.3390/jcm12185988

**Published:** 2023-09-15

**Authors:** Janusz Pach, Piotr A. Regulski, Jacek Tomczyk, Jerzy Reymond, Katarzyna Osipowicz, Izabela Strużycka

**Affiliations:** 1Department of Comprehensive Dentistry, Medical University of Warsaw, Binieckiego 6 St., 02-097 Warsaw, Poland; biuro@megadental.info.pl (J.P.); istruzycka@gmail.com (I.S.); 2Laboratory of Digital Imaging and Virtual Reality, Department of Dental Radiology and Maxillofacial Imaging, Medical University of Warsaw, Binieckiego 6 St., 02-097 Warsaw, Poland; 3Institute of Biological Sciences, Cardinal Stefan Wyszyński University, Woycickiego 1/3 St., 01-938 Warsaw, Poland; j.tomczyk@uksw.edu.pl; 4Department of Maxillofacial Surgery, Radom Specialist Hospital, Tochtermana 1 St., 26-600 Radom, Poland; reymond_j@o2.pl; 5Department of Dermatology, Medical University of Warsaw, Koszykowa 82A St., 02-008 Warsaw, Poland; osipowicz.kasia@gmail.com

**Keywords:** taurodontism, endodontic treatment, orthodontic treatment, pulp chamber, historical populations

## Abstract

Taurodontism is a morphological anomaly of multirooted molars characterized by apical displacement of the pulp chamber, shortened roots, and the absence of constriction at the dentoenamel junction. It can negatively impact the outcome of dental treatment plans. This study aimed to compare the prevalence of taurodontism among contemporary and historical populations from Radom, Poland. Five hundred eighty-two panoramic radiographs of contemporary patients and 600 radiographs of historical individuals were analyzed using the Shifman and Chanannel index. Group differences were determined with Pearson’s chi-square tests according to sex, site, tooth group, and historical period. The study also evaluated the degree of severity of taurodontism in relation to dental groups, gender, and the periods from which contemporary patients as well as historical individuals originated. In the contemporary population, taurodontism was observed in 34% of individuals. In the historical data, the highest prevalence of taurodontism (31%) was observed among individuals from the 18th and 19th centuries, while earlier periods exhibited considerably lower prevalence rates. Across contemporary and historical populations, the maxillary molars were the most commonly affected teeth. Hypotaurodontism was the most prevalent form of taurodontism. The prevalence of taurodontism has gradually increased from the 11th century to the current day. The results of the research are of great importance for the clinician in terms of planning comprehensive dental treatment.

## 1. Introduction

Taurodontism is a morphological anomaly that affects multirooted molars in the mandible and maxilla. It has been observed in both permanent and deciduous teeth [1,2,3]. The first report of these atypically shaped teeth in prehistoric hominids was made by deTerra in 1903 [4,5]. In 1908, Dragutin Gorjanovic-Kramberger was the first to describe taurodontism based on a pre-Neanderthal fossil found in Krapina, Croatia [5]. The presence of taurodontism in early hominids offers invaluable insights into human evolution. Through the examination of the prevalence of this anomaly, the migratory patterns of populations can be determined. Elucidating such patterns can provide a richer understanding of the demographic history and cultural interactions of human and hominid populations over time. The historical context of taurodontism is therefore not only a record of a dental anomaly in ancient hominids but also a window into human evolutionary history. Through the study of taurodontism, we can gain a deeper understanding of the genetic, dietary, and environmental factors that have shaped human dentition over millennia.

The presence of taurodontism in modern humans was first noted by Henry Pickerill in 1909 [3]. The term “taurodontism” was coined in 1913 by Sir Arthur Keith, derived from the Latin word “taurus” (bull) and the Greek word “odus” (tooth), i.e., a “bull-like tooth”. Keith also used the term “cynodont” to describe teeth without this anomaly, characterized by a proper ratio of pulp chamber height to root length [4,6].

Teeth affected by taurodontism display distinct features such as a vertically elongated pulp chamber, lack of narrowing at the cementoenamel junction (CEJ), and shortened roots. With the naked eye, these teeth appear indistinguishable from those with a typical morphology, known as cynodontic teeth. Therefore, the diagnosis of taurodontism relies on the use of radiological imaging techniques [4,7].

Dentists should be vigilant given the relative frequency of molars exhibiting this altered morphology. A thorough analysis of radiological imaging is crucial before initiating treatment. An incorrect or missed diagnosis can result in serious complications from dental procedures in both the short and long term [3].

One such dental procedure that can lead to serious complications in taurodontic teeth is root canal treatment, which poses challenges due to the complex root canal anatomy. The elongated pulp chamber and shortened roots complicate the localization and thorough cleaning of the root canal system. Such teeth may be more susceptible to pulpitis, pulp necrosis, and periodontal issues [6]. The enlarged pulp chamber renders the tooth more susceptible to infections and inflammation, which may necessitate extensive treatment or extraction [4]. Restorative procedures such as dental fillings, crowns, or other treatments can be problematic for taurodontic teeth [3]. The altered tooth shape and irregular root anatomy might compromise the stability and retention of restorations; thus, meticulous treatment plans are needed. Furthermore, taurodontism may present difficulties during orthodontic treatment, as abnormal tooth morphology and root structure can affect the alignment and stability of the teeth, potentially impacting the overall outcome of the treatment. Careful anchorage planning and (possibly) a slower rate of tooth movement may be warranted to minimize the risk of root resorption or other adverse outcomes [3,4]. To manage these clinical challenges, dentists and specialists might need to modify their treatment strategies, such as employing advanced imaging techniques, utilizing specialized endodontic instruments, or contemplating alternative restorative options.

The aetiology of taurodontism remains unclear. Many scholars have suggested that abnormal invagination of Hertwig’s epithelial sheath during early embryonic development could be a potential cause [3,5,8,9]. Taurodontism can manifest as an isolated anomaly or in conjunction with other (primarily genetic) conditions, such as Down syndrome, Klinefelter syndrome, or tricho-dento-osseous syndrome [10,11,12]. Some studies have reported a correlation between genes on the X chromosome and the prevalence of this anomaly. The X chromosome gene responsible for enamel development may also play a role in taurodontism [13,14]. Diagnosing this dental anomaly can lead to the early detection of various syndromes [15], as taurodontism exhibits a polygenic inheritance pattern governed by a few genes, with at least one gene located on the X chromosome [16]. The occurrence of taurodontism may also be associated with environmental factor. Within the existing literature, a case is described involving a young patient who endured bacterial osteomyelitis affecting the jawbone and marrow. Subsequent to this medical episode, the individual was diagnosed with taurodontism in the third molar tooth that erupted at a later time. The researchers behind the study postulated that the onset of this dental anomaly could be attributed to an environmental factor, specifically bacterial influence, rather than being solely conditioned by genetic variables. This case prompts a reconsideration of the etiological factors contributing to taurodontism, suggesting that environmental factors, such as bacterial infections, may play a more significant role than previously assumed. [17].

The aim of this study was to compare the prevalence of taurodontism in the contemporary population near Radom, Poland with the prevalence of this anomaly in historical individuals from the 11th to the 19th centuries from the same area. If taurodontism is genetically conditioned, then one would anticipate observing variability between contemporary and historical populations. Conversely, the null hypothesis posits that there is no significant effect between different populations.

## 2. Materials and Methods

### 2.1. Contemporary Population

First, 582 panoramic radiographs obtained in 2022 from patients at the Radom Specialist Hospital using a stationary CS 8000C panoramic X-ray unit (Carestream Health, Toronto, ON, Canada) were inspected. For inclusion, the radiographs had to be of high quality, free from artefacts related to improper patient positioning or exposure conditions, and had to display at least one intact multirooted molar without any prior endodontic treatment or prosthetic restoration. Conversely, the exclusion criteria were as follows: radiographs of poor quality; those that did not contain at least one multirooted molar; or those that exclusively contained teeth with prior endodontic treatment, extensive decay, or prosthetic restorations. Based on these criteria, 179 radiographs were excluded, with the remaining 403 radiographs, comprising 2198 multirooted molars, included in the study (see Figure 1). The study encompassed a total of 923 multirooted molars from female participants and 1275 multirooted molars from male participants. The average age of the individuals in the study was 39 years, with a standard deviation of 19 years. The second mandibular and maxillary molars were the most common teeth assessed, each accounting for 22% of the radiographs, while the third mandibular and maxillary molars were the least common, accounting for 13% and 11%, respectively. Ethical approval for conducting the research was obtained from the local bioethics committee under reference number AKBE/135/2023. The available research material did not contain sensitive patient data, such as names or medical history. The nature of the current study was retrospective.

### 2.2. Historical Population

In the historical population analysis, 600 dental, periapical radiographs from individuals who spanned the 11th to the 19th centuries and lived in Radom, Poland were assessed. The specimens originated from the archaeological excavations in Radom. These individuals were further categorized into three historical periods: those from the early Middle Ages (11th–12th centuries), the late Middle Ages (14th–17th centuries), and the modern period (18th–19th centuries). These radiographs were captured using a portable X-ray unit (EZX-60, Edlen Imaging, Scottsdale, AZ, USA), adhering to paralleling technique. The employment of a portable device afforded us the flexibility to obtain images even in instances where fragments of the skull were absent. The exclusion criteria were the same as those for the analysis of panoramic radiographs; additionally, images were excluded if the sex of the individual could not be identified. Consequently, 349 radiographs were excluded, with the remaining 251 dental radiographs (comprising 640 multirooted molars) included in further analysis; of these, 258 multirooted molars belonged to female individuals, and 382 belonged to male individuals. Most of these molars were second and first lower molars, comprising 36% and 29% of the molars, respectively, while the second and third molars accounted for the fewest teeth, at 6% and 4%, respectively (see Figure 2).

### 2.3. Measurements

The method used in this study was based on the Schiffman and Chanannel taurodont index (*TI*), as described in 1978 [3,18]. This index was employed because it is a well-established and widely accepted method for quantifying taurodontism, providing a standardized means of measuring the pulp chamber and root dimensions. Its use ensures consistency and comparability of the results with previous studies, and this index can be used with both panoramic and periapical radiographs. The *TI* can be calculated from the ratio of the distance between the lowest point of the pulp chamber roof and the highest point of its floor (*a*) to the distance from the lowest point of the pulp chamber roof to the apex of the longest root of the tooth (*b*). This ratio is then multiplied by 100, as per the formula below:$$TI=\frac{a}{b}\cdot 100$$

According to the adopted scale, taurodontic teeth are classified as follows: hypotaurodontic (*TI* values of 20.0 to 29.9), mesotaurodontic (*TI* values of 30.0 to 39.9), or hypertaurodontic (*TI* values of 40.0 to 75.0). These values are most frequently used in the studies described in the available literature for the assessment of the degree of taurodontism [3,4,15].

### 2.4. Statistical Analysis

Biometric measurements of multirooted molars were collected using the MicroDicom v.2023.1 application (Sofia, Bulgaria). The radiographs were evaluated by two experienced dentists specializing in dental and maxillofacial radiology, and the interrater reliability was assessed using the interclass correlation coefficient. The Pearson chi-square test for independence was used to compare the prevalence of taurodontism and its severity between independent groups according to sex, site, tooth group, and historical period, with a significance level set at 0.05.

## 3. Results

### 3.1. Contemporary Population

Taurodontism was diagnosed in 34% of all the teeth of contemporary subjects (750/2198), with a notable prevalence in 31% of male teeth and 38% of female teeth; this sex difference was statistically significant (*p* = 0.0014). The severity of taurodontism was consistent between sexes. Among the taurodontic teeth, 77% were hypotaurodontic teeth, 18% were mesotaurodontic teeth, and 5% were hypertaurodontic teeth (Table 1, Figure 3).

The biometric analysis revealed that taurodontism predominantly affected maxillary teeth (52% of upper teeth), with only 16% of mandibular teeth affected (*p* < 0.0001). Both sexes exhibited this trend. The anomaly was slightly more pronounced on the right side (54%) than on the left side (50%), but this difference was not statistically significant (Table 2).

Among the taurodontic teeth, third upper molars (78%) accounted for the most teeth, and first mandibular molars (4%) accounted for the fewest teeth. A statistically significant difference in the frequency of taurodontism was observed among the different tooth groups (*p* < 0.0001). First lower molars had the highest frequency of hypotaurodontism (92%), while third upper molars had the lowest frequency of hypotaurodontism (56%). Mesotaurodontism and hypertaurodontism were most prevalent in third upper molars (33% and 11%, respectively) (Table 3, Figure 3). Sample panoramic radiographs with taurodontic teeth are presented in Figure 4.

### 3.2. Historical Population

In the historical populations, taurodontism was found in 27% of teeth (176/640), with no significant sex difference in prevalence rates. Taurodontism was identified in 53% of maxillary teeth, predominantly in second and first molars, and in 22% of mandibular teeth. Hypotaurodontism was the most common form of taurodontism (81%), followed by mesotaurodontism (18%) and hypertaurodontism (1%). The prevalence of taurodontism varied significantly across historical periods (*p* = 0.003). Representative periapical radiographs are presented in Figure 5.

### 3.3. Comparison of Prevalence of Taurodontism between Populations

Among both contemporary and historical populations, taurodontism was observed in 32% of teeth. An evaluation of the severity of taurodontism across all periods revealed that hypotaurodontism was the most common form, accounting for 78% of taurodontic teeth. Mesotaurodontic teeth accounted for 18% of taurodontic teeth, whereas hypertaurodontic teeth were rare, accounting for only 4%. There was a significant difference in the prevalence of taurodontism across chronological periods (*p* = 0.0030), with a trend towards increased prevalence in more recent periods (Table 4).

The evaluation of methodological error indicated that the method was highly reliable. The ICC value (0.91) suggests that the measurements introduced a negligible amount of error during the assessments.

## 4. Discussion

Taurodontism, a developmental anomaly manifesting in multirooted molar teeth, is often observed within various populations, although its prevalence is debated [18]. While some authors [3,4,10] regard this condition as an uncommon morphological aberration, studies have shown that the prevalence of taurodontism is surprisingly variable. For example, some studies have suggested that this anomaly appears in 0.25% to 11.3% of individuals [3]. Notably, data from the Chinese population has indicated that taurodontism might be observed in up to 46% of individuals [4,19]. In contrast, investigations in Germany indicate a low prevalence rate of 2% [20], while research from Israel observed a prevalence of 11.5% in examined teeth [18].

Both contemporary and historical populations, as described in this research, have consistently reported a higher prevalence of taurodontism in maxillary molars compared to their mandibular counterparts, a finding echoed by other scholars [4,8,11,18,21]. Interestingly, a study in the Turkish population suggested a greater prevalence of taurodontism in mandibular molars [19].

It is essential for dental practitioners to be keenly aware of the far-reaching consequences of taurodontism on the full course of dental treatment. This study emphasizes the need to conduct an exhaustive biometric evaluation of radiographic data at the practitioner’s disposal. An accurate diagnosis is crucial for consideration of therapeutic interventions. In particular, the performance of endodontic procedures on taurodontic teeth (chiefly mesotaurodontic and hypertaurodontic teeth) is complex and requires greater vigilance and engagement from the dentist. The elongation of the pulp chamber and concomitant shortening of the root canal hinder identification of the location of root canal openings or apical foramina, increasing the risk of perforation. The utilization of magnification tools, such as dental operating microscopes, and enhanced obturation techniques has the potential to vastly improve the success rates of root canal treatments in taurodontic teeth. Surgical, prosthetic, and orthodontic procedures should also be approached with caution due to the potential complications, such as root fractures owing to the proclivity of root bifurcation in taurodontic teeth [3]. Furthermore, the shorter root length of these teeth weakens the anchorage within the alveolar socket, rendering them unsuitable as abutment teeth in prosthetic applications [4]. Additionally, contraindications for orthodontic treatments are common due to the predilection for root resorption of taurodontic teeth [22]. Considering the relative prevalence of this morphological anomaly in multirooted molars within modern populations, as evidenced by this study, clinicians should be diligent in examining radiological images for taurodontism. Such an approach is critical for ensuring optimal outcomes and preventing complications in therapeutic interventions. The combination of advanced dental technology, evidence-based practice, and clinical acumen can pave the way for a more enlightened and efficacious approach to managing taurodontism. The findings of our study substantiate a growing prevalence of taurodontism over time. Based on these empirical results, one can reasonably project that taurodontism is poised to become an increasingly formidable challenge in the realm of contemporary dentistry. The rising trend suggests that dental practitioners will likely encounter cases of taurodontism with greater frequency, requiring enhanced diagnostic protocols and potentially specialized treatment approaches to manage this condition effectively.

The research described in this manuscript, which examined historical samples from the 11th to 19th century from the Radom region, revealed a variation in the prevalence of taurodontism across historical periods. The demographic landscape of Radom underwent substantial alterations after the 14th century, largely due to a surge in migration as individuals flocked to the area in pursuit of employment opportunities and an improved standard of living. Furthermore, the 18th and 19th centuries brought additional changes to the population composition, spurred by economic and social transformations. This migratory trend ceased in the 19th century and has remained negligible into the 21st century. Notably, the prevalence of taurodontism was conspicuously higher among individuals from the 18th/19th centuries, a development that could plausibly be ascribed to the intensified migration into Radom during that era. This migration pattern has been further substantiated by genetic analyses [23,24].

Evidence suggests that there is a potential association between taurodontism and sex [15,17,25]. The biometric assessment of panoramic radiographs obtained from contemporary residents of Radom revealed a significant sex difference in the prevalence of this anomaly. Consistent with investigations including diverse populations, taurodontism seemed to be more prevalent in females than in males [18,26]. Conversely, some studies [12] have refuted any sex difference. In the historical populations, sex did not significantly influence the prevalence of taurodontism. However, it is crucial to note the potential effect of sample size on these outcomes. The larger the sample size was, the greater the difference between sexes. The genomic basis of taurodontism, specifically the localization of the responsible gene on the X chromosome, may explain the higher incidence among females [1,13]. Further supporting this hypothesis, the gene responsible for enamel formation is located on the X chromosome. Although the precise genetic underpinnings have yet to be fully elucidated, mutations or variations in genes involved in tooth development could be contributing factors. Nevertheless, the paucity of findings hampers conclusions regarding the correlation between genetics and taurodontism. Potentially, a more thorough investigation into the genes of the X chromosome could provide deeper insight into the mechanism of the defect formation.

The severity of this anomaly did not significantly vary across contemporary or historical periods, consistent with other findings [18,21]. The gene responsible for taurodontism, linked to X chromosome aneuploidy, may contribute to differences in severity. Some researchers have suggested that the severity of taurodontism might be correlated with the presence of associated systemic disorders such as Down syndrome, Klinefelter syndrome, tricho-dento-osseus syndrome and others [1,9,13,18,25,27].

### 4.1. Limitations

One of the limitations of this study is the absence of an in-depth medical history for the participants. The data from contemporary subjects were anonymized, which (while vital for maintaining privacy) constrained the scope of analyses regarding subject characteristics (i.e., to sex only, rather than any underlying systemic diseases). This deficiency casts a shadow of uncertainty over the interplay between taurodontism and systemic diseases. Consequently, essential questions remain unanswered, highlighting the urgent need for more comprehensive investigations in this sphere.

### 4.2. Future Perspectives

One of the most compelling areas awaiting further exploration is the genetic landscape of taurodontism. A deeper understanding of the genes involved and the interplay of genetic factors could elucidate the intricacies of this condition and its development. The evolutionary aspect of taurodontism is equally enthralling; probing into the evolutionary importance and adaptive value of this condition could provide invaluable background information. Furthermore, given the clinical challenges of taurodontism, there is a pressing need to develop and refine techniques to achieve tailored dental management in individuals with this anomaly. An investigation into preventive measures and strategies for early diagnosis could change the narrative for many patients. The integration of innovative technologies, such as artificial intelligence, for the efficient detection of taurodontism through dental radiographs could be transformational.

The subject of taurodontism occurrence in historical individuals is infrequently addressed in the available literature. This research has demonstrated an ascending prevalence of taurodontism from the early Middle Ages to contemporary times. This lends credence to the notion that the manifestation of this dental anomaly may be influenced by genetic diversity, potentially induced by migrations.

## 5. Conclusions

The prevalence of taurodontism among the contemporary population from Radom exceeded rates in historical populations from the 11th/12th and 14th/17th centuries. This observation may be linked to increased migration during the 14th/17th centuries. In both contemporary and historical cohorts diagnosed with taurodontism, individuals were predominantly affected by the mildest form of the condition, hypotaurodontism. Hypertaurodontism was relatively rare. An analysis of data collected across all periods revealed a markedly higher prevalence of the anomaly in maxillary rather than mandibular teeth. Among the contemporary population, the condition was more prevalent in females than in males, however in the historical population, there were not sex differences in taurodontism severity.

The insights gleaned from this research can help clinicians to formulate interdisciplinary dental treatment strategies that account for the various facets of taurodontism.

## Figures and Tables

**Figure 1 jcm-12-05988-f001:**
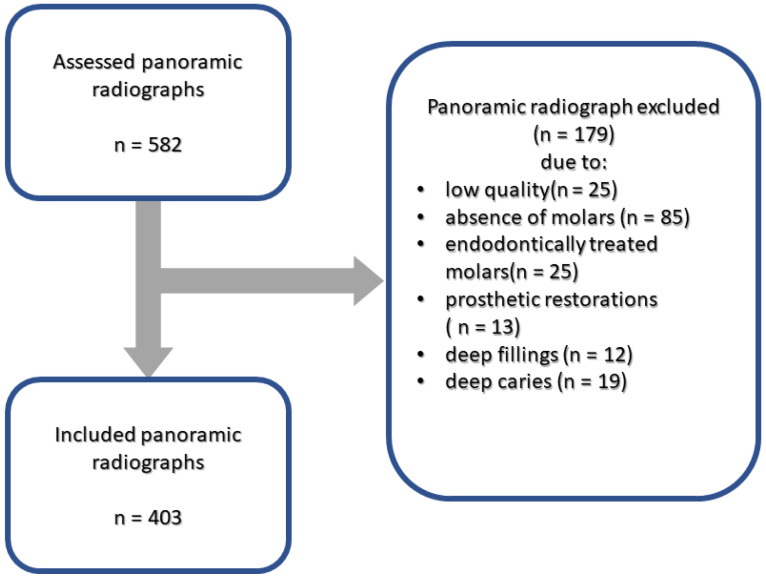
Flow chart of the selection of panoramic radiographs from the contemporary population.

**Figure 2 jcm-12-05988-f002:**
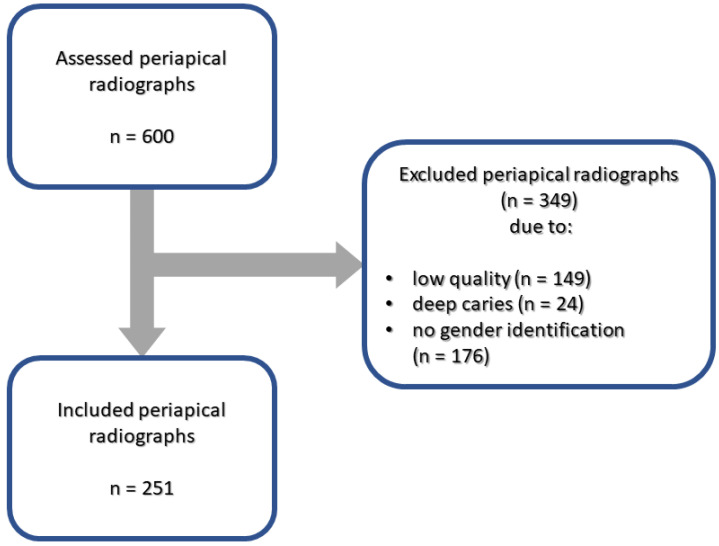
Flow chart of the selection of periapical radiographs from the historical populations.

**Figure 3 jcm-12-05988-f003:**
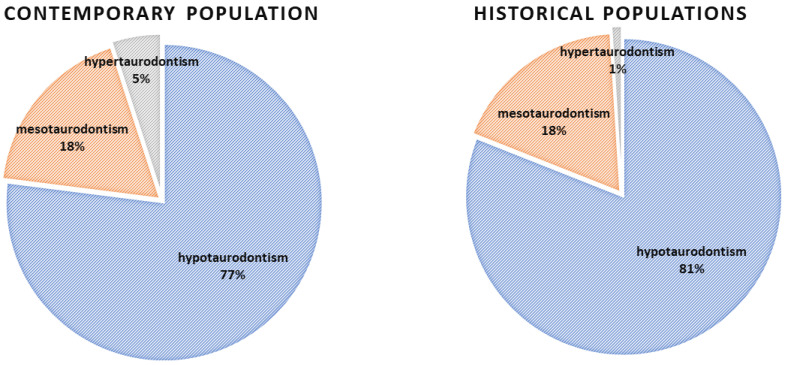
Prevalence of severity of taurodontism in the contemporary and historical populations.

**Figure 4 jcm-12-05988-f004:**
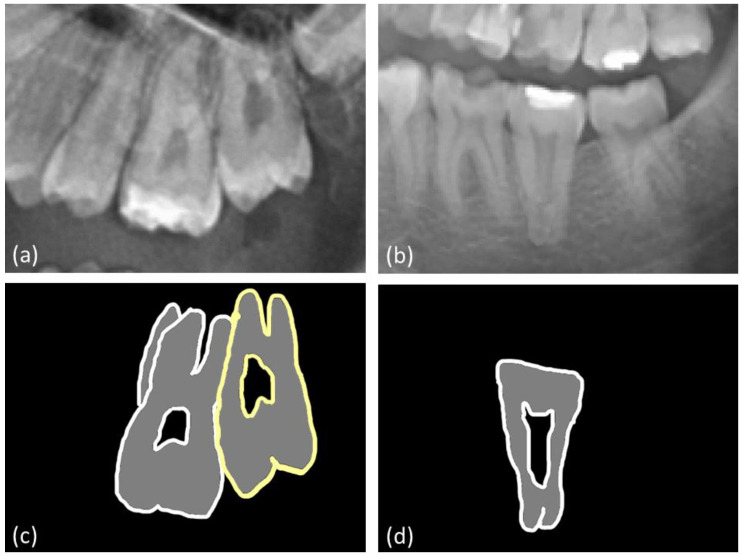
Representative panoramic radiographs of (**a**) hypotaurodontic, mesotaurodontic, and (**b**) hypertaurodontic teeth, with corresponding outlines shown below; (**c**,**d**).

**Figure 5 jcm-12-05988-f005:**
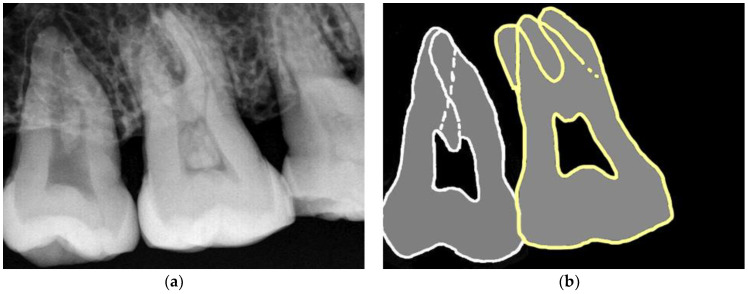
Representative radiographs of (**a**) hypotaurodontic, mesotaurodontic, (**b**) as well as their outlines.

**Table 1 jcm-12-05988-t001:** Frequency of taurodontic teeth according to sex and severity in the contemporary population.

Sex	Taurodontic Teeth	Hypotaurodontic Teeth	Mesotaurodontic Teeth	Hypertaurodontic Teeth	*p* Value (Taurodontism to Sex)
F	350/923 (38%)	272/350 (78%)	59/350 (17%)	19/350 (5%)	0.0014
M	400/1275 (31%)	307/400 (77%)	73/400 (18%)	20/400 (5%)	

**Table 2 jcm-12-05988-t002:** Frequency of taurodontism in upper, lower, right and left molars of the contemporary population.

Tooth Group	Taurodontic Teeth	Cynodontic Teeth	*p* Value
Upper molars	583	539	<0.0001
Lower molars	167	909	
Right molars	396	726	0.2366
Left molars	354	722	

**Table 3 jcm-12-05988-t003:** Frequency of taurodontism among tooth groups according to severity in the contemporary population.

Tooth Group	Taurodontic Teeth	Hypotaurodontic Teeth	Mesotaurodontic teeth	Hypertaurodontic Teeth	*p* Value (Affected Teeth in Each Tooth Group)
First upper molars	128/382 (33%)	115/128 (90%)	12/128 (9%)	1/128 (1%)	0.0001
Second upper molars	250/477 (52%)	206/250 (82%)	34/250 (14%)	10/250 (4%)	
Third upper molars	205/263 (78%)	115/205 (56%)	67/205 (33%)	23/205 (11%)	
First lower molars	13/320 (4%)	12/13 (92%)	1/13 (8%)	0/13 (0%)	
Second lower molars	56/483 (12%)	47/56 (84%)	7/56 12%)	2/56 (4%)	
Third lower molars	98/273 (36%)	84/98 (86%)	11/98 (11%)	3/98 (3%)	

**Table 4 jcm-12-05988-t004:** Differences in the prevalence of taurodontism in contemporary and historical populations.

Time Period	Taurodontic Teeth	Hypotaurodontic Teeth	Mesotaurodontic Teeth	Hypertaurodontic Teeth	*p* Value (Prevalence of Taurodontism across Historical Periods)
Early Middle Ages	26/120 (22%)	21/26 (81%)	5/26 (19%)	0/26 (0%)	*p* = 0.0030
Late Middle Ages	12/76 (16%)	9/12 (76%)	3/12 (25%)	0/12 (0%)	
Modern period (18th–19th centuries)	138/444 (31%)	114/138 (83%)	22/138 (16%)	2/138 (1%)	
Contemporary period (2022)	750/2198 (34%)	579/750 (77%)	132/750 (18%)	34/750 (5%)	
Total	926/2838 (32%)	723/926 (78%)	162/926 (18%)	36/926 (4%)	

## Data Availability

The data that support the findings of this study are available from the corresponding author upon reasonable request.

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
