# Peer review of "Prevalence of Taurodontism in Contemporary and Historical Populations from Radom: A Biometric Analysis of Radiological Data"

_jcm, 2023, doi:10.3390/jcm12185988_

Round 1

Reviewer 1 Report

I am afraid to say the design of the research was not proper. The X-ray was first discovered in 1895. The first dental radiograph was made in 1896. How could the authors use X-day to analyze the historical population of individuals who spanned the 11th to the 19th centuries and lived in Radom, Poland? Because from the 11th to 17 centuries, there was no X-ray invention. Although the structure of this article showed scientific information, I personally doubt the truth of the results.

The English editing is well, but the truth of the article was not confident. I highly recommend the authors explain the experimental of the "historical group" because the dental X-ray was invented in 1896. Before 1896, how could the authors collect periapical film from residents who lived in Random, Poland in the 11th to 18th centuries? 

Author Response

Thank you for your review. We would like to clarify that we performed X-ray images of the teeth of historical individuals in the years 2021-2022, using skulls obtained from excavations in Radom.  This has been described in the text: “In the historical population analysis, 600 dental radiographs from individuals who spanned the 11th to the 19th centuries and lived in Radom, Poland were assessed. The specimens originated from the archaeological excavations in Radom. These individuals were further categorized into three historical periods: those from the early Middle Ages (11th-12th centuries), the late Middle Ages (14th-17th centuries), and the modern period (18th-19th centuries). These radiographs were captured using a portable X-ray unit (EZX-60, Edlen Imaging, USA).”

Reviewer 2 Report

Dear authors,

Thank you for submitting your valuable work to the journal. The topic of the research is intersting and could bring valuable contributions to the subject. However, there are some changes I would suggest in order to increase the manuscript's scientific accuracy:

- Please add a Null Hypothesis to the Objectives of your study

- Please describe in more detail the x-raying procedure of the hystoric population (how were the panoramic xrays performed on the assessed skulls)

- Please discuss on the possible evolution of taurodontism's prevalence from Early Middle Ages to Modern Era

- Please discuss on Clinical implications of your findings

We look forward to receiving the revised version of your manuscript.

Kind regards

minor check-up of English grammar and vocabulary

Author Response

Thank you for your comprehensive review and valuable suggestions aimed at enhancing the scientific accuracy and impact of our manuscript. We have carefully addressed each of your points as follows:

- Please add a Null Hypothesis to the Objectives of your study

We have incorporated a null hypothesis into the objectives section of our study:  “If taurodontism is genetically conditioned, then one would anticipate observing variability between contemporary and historical populations. Consequently, the null hypothesis posits that there is no significant effect between different populations.”

- Please describe in more detail the x-raying procedure of the hystoric population (how were the panoramic xrays performed on the assessed skulls)

In the historical population the dental, periapical radiographs were taken (not panoramic radiographs). Specifically, periapical radiographs were captured using a portable X-ray unit and adhering to the paralleling technique. This portable device provided the flexibility to capture images even when skull fragments were absent. This information has been duly integrated into the revised manuscript.

- Please discuss on the possible evolution of taurodontism's prevalence from Early Middle Ages to Modern Era

According to the Reviewer’s suggestion, the following discussion has been added to the manuscript: “The subject of taurodontism occurrence in historical individuals is infrequently ad-dressed in the available literature. This research has demonstrated an ascending prevalence of taurodontism from the Early Middle Ages to contemporary times. This lends credence to the notion that the manifestation of this dental anomaly may be influenced by genetic diversity, potentially induced by migrations.”

- Please discuss on Clinical implications of your findings

In accordance with the Reviewer's suggestion, the paragraph pertaining to clinical implications has been revised and updated: “It is essential for dental practitioners to be keenly aware of the far-reaching consequences of taurodontism on the full course of dental treatment. This study emphasises the need to conduct an exhaustive biometric evaluation of radiographic data at the practitioner’s disposal. An accurate diagnosis is crucial for consideration of therapeutic interventions. In particular, the performance of endodontic procedures on taurodontic teeth (chiefly mesotaurodontic and hypertaurodontic teeth) is complex and requires greater vigilance and engagement from the dentist. The elongation of the pulp chamber and concomitant shortening of the root canal hinder identification of the location of root canal openings or apical foramina, increasing the risk of perforation. The utilization of magnification tools, such as dental operating microscopes, and enhanced obturation techniques has the potential to vastly improve the success rates of root canal treatments in taurodontic teeth. Surgical, prosthetic, and orthodontic procedures should also be approached with caution due to the potential complications, such as root fractures owing to the proclivity of root bifurcation in taurodontic teeth [3]. Furthermore, the shorter root length of these teeth weakens anchorage within the alveolar socket, rendering them unsuitable as abutment teeth in prosthetic applications [4]. Additionally, contraindications for orthodontic treatments are common due to the predilection for root resorption of taurodontic teeth [22]. Considering the relative prevalence of this morphological anomaly in multirooted molars within modern populations, as evidenced by this study, clinicians should be diligent in examining radiological images for taurodontism. Such an approach is critical for ensuring optimal outcomes and preventing complications in therapeutic interventions. The combi-nation of advanced dental technology, evidence-based practice, and clinical acumen can pave the way for a more enlightened and efficacious approach to managing taurodontism. The findings of our study substantiate a growing prevalence of taurodontism over time. Based on these empirical results, one can reasonably project that taurodontism is poised to become an increasingly formidable challenge in the realm of contemporary dentistry. The rising trend suggests that dental practitioners will likely encounter cases of taurodontism with greater frequency, requiring enhanced diagnostic protocols and potentially specialized treatment approaches to manage this condition effectively.

We trust that these revisions address your concerns effectively and contribute to the scientific rigor and applicability of our research.

Thank you again for your constructive feedback.

Reviewer 3 Report

Dear authors, the manuscript is very interesting but I suggest you explain some parts better:

line 97 :what could be the environmental factors ?please list some of them.

line 170-171: I suggest you make a pie chart related to this data.

line 219 :cite these "authors" referred to

line 227: cite these "both contemporary and historical investigations".

line 271 :cite "some studies"

divide the limitation part of the study, in my opinion it should be a separate part from the discussion and also include the part of future perspectives about the diagnosis of this disease as a separate part from the discussion.

Please improve the manuscript so that it may be more suitable 

English needs to be improved to be more understandable English needs to be improved to be more understandable

Author Response

Thank you for your comprehensive review and valuable suggestions aimed at enhancing the scientific accuracy and impact of our manuscript. We have carefully addressed each of your points as follows:

line 97 :what could be the environmental factors ?please list some of them.

The author whom we have cited (Sears, J. "Taurodontism in Modern Populations." Dent. Anthropol. J. 2018, 14, 14–19; DOI:10.26575/daj.v14i2.185) referenced a case involving a patient who, during adolescence, experienced bacterial inflammation of the jawbone and marrow. Subsequently, taurodontism was identified in a later-erupting third molar. The authors of the study concluded that the manifestation of this dental anomaly could be attributed to an environmental factor (bacterial), as opposed to genetic causes.

line 170-171: I suggest you make a pie chart related to this data.

In accordance with the Reviewer's suggestion, a pie chart has been added to represent the data.

line 219 :cite these "authors" referred to

As per the Reviewer's suggestion, the authors have now been cited

line 227: cite these "both contemporary and historical investigations".

In response to the Reviewer's suggestion for clarification, the sentence has been revised to unambiguously convey its meaning. The amended sentence now reads: "Both contemporary and historical populations, as described in this research, have consistently reported a higher prevalence of taurodontism in maxillary molars compared to their mandibular counterparts, a finding echoed by other scholars [4,8,11,18,21]."

line 271 :cite "some studies"

As per the Reviewer's recommendation, the necessary citation has been added.

divide the limitation part of the study, in my opinion it should be a separate part from the discussion and also include the part of future perspectives about the diagnosis of this disease as a separate part from the discussion.

In accordance with the Reviewer's recommendations, we have separated the "Limitations" section from the main discussion and have also included a distinct section for "Future Perspectives". We greatly appreciate your insightful suggestions, which have been invaluable in enhancing both the clarity and depth of our manuscript.

Round 2

Reviewer 1 Report

I made a big mistake with my previous questions. I wrote to the editor and finally got the chance to collect my misunderstanding of your research. This research was good for publication. I admired your contribution to dental science. Congratulations!!

Reviewer 3 Report

The authors followed my advice and improved the quality of the manuscript.

I do not ask for any further changes.

Congratulations to the authors.